# Health Behaviors and Self-Reported Oral Health among Centenarians in Nanjing, China: A Cross-Sectional Study

**DOI:** 10.3390/ijerph18147285

**Published:** 2021-07-07

**Authors:** Xin Xu, Yuan Zhao, Danan Gu, Yaolin Pei, Bei Wu

**Affiliations:** 1Population Research Institute, Nanjing University of Posts and Telecommunications, Nanjing 210042, China; xuxin199010@126.com; 2Ginling College, Nanjing Normal University, Nanjing 210023, China; 3Independent Researcher, New York, NY 10010, USA; gudanan@yahoo.com; 4Rory Meyers College of Nursing, New York University, New York, NY 10010, USA; yp22@nyu.edu

**Keywords:** self-reported oral health, health behaviors, education, living arrangements, Chinese centenarians

## Abstract

The role of health behaviors in oral health conditions in individuals of extremely old age remains understudied. This study included 185 participants aged 100 years or older from the Nanjing Centenarians Study (NCS) to examine the associations between health behaviors and oral health and investigate the potential moderating role of education and living arrangements in such relationships. The oral health status as an outcome included the self-reported oral health status and edentulous status. Health behavior variables included smoking, eating fruits, eating vegetables, participating in leisure activities, and practicing oral hygiene behaviors. Sociodemographic characteristics and health status were considered as confounders. Descriptive statistics, ordinal regression, and logistic regression models were used to address the research questions. Results showed that better oral health was reported by centenarians who were non-smokers, participated in more leisure activities, and practiced higher frequency of oral hygiene behaviors. Those who ate fruits daily and practiced more frequently oral hygiene behaviors were more likely to be dentate. The positive association of oral hygiene behaviors was stronger for centenarians who were formally educated and co-resided with family members. The results suggest that effective interventions should consider health behaviors and living arrangements in this growing population to improve their oral health status.

## 1. Introduction

Worldwide, the number of centenarians is projected to increase six-fold from 2019 to 2050 [1]. China is no exception; the number of self-claimed centenarians rose dramatically from 3851 in 1982 to 35,934 in 2010 [2,3]. Oral health is a critical component of healthy aging [4,5], and it affects the general health and well-being of individuals throughout their life course [6,7]. Individuals with poor oral health, such as tooth loss, may have trouble eating and swallowing, have inadequate nutritional intake, and experience increased struggles to speak and smile effectively. These consequences all affect their general health, quality of life, and longevity [8,9]. Thus, improving oral health among this rapidly increasing population group has become a priority for public health.

Health behaviors, such as abstaining from smoking [10,11,12,13], practicing a healthy diet [14,15,16,17,18], performing adequate physical activities [19,20,21,22,23,24], and engaging in proper hygiene practices [25], are essential determinants of longevity [26] and could also influence oral health in later life [27,28]. Poor oral hygiene behavior is among the most critical factors related to oral health and oral health problems (e.g., periodontitis, dental caries, gum pain and infection, and tooth loss) [29]. Toothbrushing with fluoride toothpaste is considered one of the most effective measures to prevent dental caries because it could eliminate the pathogenic factors associated with dental caries, gingivitis, and periodontal disease [30,31,32,33,34]. However, no study has focused on the association between oral hygiene behaviors and oral health among Chinese centenarians. Centenarians may also have different oral hygiene behaviors compared with other groups of people. Many may decrease toothbrushing frequency or lose the ability to perform oral hygiene independently because of functional and cognitive impairment [35,36,37]. However, some may maintain this lifelong oral hygiene practice. The oral hygiene behavior and oral health among centenarians is unknown, particularly among Chinese centenarians. More research is needed in this area to promote oral health and healthy aging in this population. Notably, the association between the oral hygiene behavior and oral health may depend on some contextual factors. Given that the protective roles of education and living arrangements have been documented in various health studies [38,39,40,41,42,43], we speculate that education and living arrangements may moderate the association between the oral hygiene behavior and oral health.

Furthermore, the oral health issues of Chinese older adults have not attracted considerable attention in Chinese society, partially due to China’s unique healthcare system, socioeconomic development, and cultural background. The yearly share of allocated oral health resources of China’s total public health funding is much smaller than the 5–10% share in developed countries [44]. Although no official statistics concerning oral health care expenses in China exist, specialized dental care is generally unavailable and unaffordable for many people because of limited insurance coverage [45]. For older adults, dental care is challenging due to their lower socioeconomic status, disadvantageous physical health, and socio-cultural reasons. On the one hand, limited mobility and financial ability of older adults reduce their access to dental care, and the current dental care system is undeveloped, resulting in the lack of dental care services and nursing staff in rural areas, townships, and communities [46,47]. On the other hand, Chinese older adults are less likely to use preventative dental utilization owing to traditional beliefs and a lack of oral health knowledge [48]. For example, most Chinese older adults believe that internal fire (which signals an important body energy imbalance in traditional Chinese medicine) is the cause of gum swelling and bleeding [49]. They believe gum bleeding and tooth loss are normal, and they tend to use traditional treatments, such as resting, avoiding stress, limiting hot food intake, and taking traditional herbal remedies [49]. These behaviors reduce individuals’ utilization of preventative dental services [50]. Given the above socioeconomic and cultural context, research on the oral health of Chinese older adults should be prioritized.

Using the data from the Nanjing Centenarians Study (NCS), the objectives of this study are to (i) describe the oral health status of centenarians in China, (ii) examine the associations between health behaviors and oral health among centenarians, and (iii) explore the moderating role of education and living arrangements in this association.

## 2. Materials and Methods

### 2.1. Sampling and Recruitment

The Nanjing Centenarian Study (NCS) was a cross-sectional study of individuals aged 100 years or older (referred to as centenarians). Specifically, most questions used in the NCS were derived from the Chinese Longitudinal Healthy Longevity Survey (CLHLS) [51], the Hong Kong Centenarian Studies [42], the Fordham Centenarian Studies (New York) [26], and several other major aging surveys in mainland China and the United States [52,53]. The questions included demographic characteristics, socioeconomic status, physical functions, cognitive function, disease conditions, psychological characteristics, lifestyles, family and social support, family history, and global deterioration assessment. All questions, particularly those with psychometrical scales, have been validated and widely used in the literature [26,42,51]. For the NCS, both quantitative and qualitative data were collected. This study was based on quantitative data because quantitative analysis was considered the most appropriate method to test and explain an association [54].

Before starting data collection, several pre-surveys were conducted from November 2017 to March 2018 to examine the adequacy of item wording to avoid any ambiguity or omission of the items used in the interviews. The formal data collection period was from June 2018 to September 2018. The inclusion criteria were as follows: (1) lived in Nanjing city at the time of the interview and (2) were aged 100 years or older by 30 April 2018, based on the household registration (hukou system). Finally, 275 older adults were identified in the registration system. Based on the list of 275 centenarians, the NCS attempted to recruit all living centenarians. During the recruitment process, the research staff visited the participants’ homes and conducted face-to-face interviews after a written informed consent was obtained, and one interview was conducted for each centenarian. During the data collection period, 42 individuals lost contact because of their death or migration and 48 individuals refused to participate due to poor health and/or severe cognitive impairment. The final sample included 185 participants (67.3% of the total identified) in this study. This sample size is larger than many centenarian studies and is sufficient for robust analyses [55,56].

In this study, all procedures involving human participants followed the ethical standards of the Institutional Review Board (or Ethics Committee) of Ginling College, Nanjing Normal University (project code: IRB-2017KYNCS001; approval date: 8 September 2017) and/or national research committee and the 1964 Helsinki Declaration and its later amendments or comparable ethical standards. Each participant or his/her legally authorized representative provided written informed consent before any data were collected.

### 2.2. Measurements

#### 2.2.1. Oral Health Outcomes

The outcome variables included the self-reported oral health status and edentulous status; these two measures capture a subjective perception and objective condition of oral health for centenarians. Similar to previous studies [49], the response to the self-reported oral health status was coded as an ordinal variable (Appendix A). Following a similar approach in previous studies [57], the edentulous status was coded as a binary variable.

#### 2.2.2. Health Behaviors and Confounders

Health behaviors included smoking, eating fruits, eating vegetables, leisure activities, and oral hygiene behaviors. Because variables such as the frequency of eating fruits and vegetables had very small sample sizes in some response categories, we grouped the responses into two categories: 0 = not every day, 1 = every day [58]. Leisure activities were measured using a summary score of the frequency for each of the 14 recoded binary activities, including doing household chores, exercising, visiting the garden, planting flowers, reading newspapers, playing cards, watching TV, calligraphy/painting, singing and dancing, fishing, photographing, chatting, and social activities. We then re-coded this measure into three categories because of its skewed distribution: 0 = no participation, 1 = participation in 1 activity, and 2 = participation in 2 or more activities [59]. In our sample, we measured oral hygiene behaviors, including toothbrushing, brushing and cleaning dentures, and other oral hygiene care, such as wiping gum surfaces with a cloth or brushing gums. The oral hygiene behavior was coded as ordinal variable varying from 1 (never) to 4 (twice a day or more); if participants maintained any of the above behaviors, we considered them as having oral hygiene behavior. In addition, regarding dental service use, health behaviors may include the frequency and intention to visit a dentist. However, considering that Chinese older adults rarely visit a dentist [48], health behavior variables in our study did not include this measure.

Sociodemographic variables and the health status were considered potential confounders in this study. Sociodemographic variables included age, gender, education, annual household income, and co-residence. The health status included activities of daily living (ADL), cognitive function, and chronic conditions. ADL was measured using the self-reported ability to perform six daily activities (bathing, dressing, toileting, indoor transferring, continence, and eating). We classified the respondents into two groups: those needing assistance in any of the six tasks (ADL dependent) versus those needing no assistance in any of these six tasks (ADL independent). Cognitive function was measured using the validated Chinese version of the Mini-Mental State Examination (MMSE), which includes six domains of cognition (orientation, reaction, calculation, short-term memory, naming, and language) with a total score of 23. Chronic conditions were measured using a dichotomous variable reflecting whether participants had ever been told by a doctor or a physician that they had one or more of the following major chronic conditions: hypertension, diabetes mellitus, heart disease, stroke, lung disease, cancer, and arthritis [60].

### 2.3. Statistical Analysis

All the indicators related to oral health were described using means (standard deviations) and frequencies (percentages) according to the self-reported oral health status and edentulous status. Group differences were compared according to the self-rated oral health status and edentulous status using t-test or ANOVA for continuous variables and chi-square test for categorical variables. We applied ordinal regression models to analyze the associations between health behavior factors and the self-reported oral health status and logistic regression models to analyze the associations between health behavior factors and edentulous status. We reported odds ratios (ORs) and 95% confidence intervals (CIs) obtained from the model-estimated robust standard errors. In Model 1, the associations between sociodemographic variables and both oral health outcomes were assessed, with other variables as potential confounders. In Model 2, the health behavioral variables were added to Model 1 while controlling for other confounders. In Model 3, all the confounders were added to Model 2. In Model 4, the interactions of “oral hygiene behavior × education” and “oral hygiene behavior × co-residence” were both added to Model 3 for the self-reported oral health status and edentulous status, respectively. Statistical analyses were performed using Stata 14.1 software.

## 3. Results

Table 1 shows the characteristics of the study sample. Only 14% of the participants reported their oral health as good or very good, and 41.6% reported that they were dentate. The mean age of the study population was 102.0 ± 2.1, and most were women (75.1%), were widowed (93.0%), were living with family members (74.6%), and had no formal education (69.2%). The average annual household income was 31,460 yuan RMB (equivalent to $4840 in 2018).

A significantly higher percentage of centenarians were non-smokers (80.5%). Only 28.6% reported that they ate fruits every day, while 62.2% reported eating vegetables every day. Most centenarians participated in leisure activities (67.0%), and fewer than half had no daily oral hygiene care daily (40.5%). Additionally, approximately 60% of the participants were ADL dependent, and two-thirds (65.9%) of the sample reported having at least one chronic disease at the time of the survey.

Table 1 further shows that men reported better oral health than women (*p* = 0.049). Centenarians who ate vegetables (*p* = 0.002) and participated in leisure activities (*p* = 0.000) were more likely to report better oral health. Those with formal education (*p* = 0.008) and those reporting more frequently oral hygiene behaviors (*p* = 0.016) were less likely to be edentulous.

### 3.1. Self-Reported Oral Health Status

The results of ordinal regression analysis are presented in Table 2. In Model 1, centenarians with formal education were more likely to report having good oral health, but this association became insignificant after health behavioral factors were considered (Model 2). However, centenarians who participated in two or more leisure activities and who frequently practiced oral hygiene behaviors were more likely to report a good oral health status, whereas former or current smokers were less likely to perceive good oral health. No difference was observed when all the confounders were considered (Model 3). When considering the two interactions, we found that the effect of oral hygiene behaviors on self-reported oral health status was more pronounced for those with formal education (Model 4).

### 3.2. Edentulous Status

Table 3 shows the results of logistic regression analysis. In Model 1, centenarians with formal education had greater odds of being dentate than those without formal education. This association was not significant when health behavioral factors were added in Model 2. Centenarians who ate fruits every day were more likely to be dentate, whereas those who ate vegetables every day were less likely to be dentate. Those who maintained more frequent oral hygiene behaviors had a higher likelihood of being dentate. However, the association between eating fruits daily and being dentate was not significant when all the confounders were considered (Model 3). When the two interactions were considered, the association between oral hygiene behaviors and edentulous status was stronger for those who co-resided with family members (Model 4).

## 4. Discussion

This study examined the associations between health behaviors and oral health outcomes among 185 centenarians in the city of Nanjing, China. Overall, these individuals’ oral health status was not good; less than 20% rated their oral health as good or very good, and 58% had lost all their natural teeth. Health behaviors, such as non-smoking, participating in two or more leisure activities, and practicing more frequently oral hygiene behaviors, increased the likelihood of good self-reported oral health, and daily fruit consumption and more frequent oral hygiene behaviors increased the likelihood of being dentate. We also found that the protective effect of frequent oral hygiene behaviors on oral health was stronger for those who had some formal education and who were living with family members.

Our finding that smoking had a negative association with self-reported oral health agreed with the belief that smoking damages the oral immune system, changes the bacterial environment, and decreases saliva function [13]. Periodontal disease and dental caries are major factors contributing to tooth loss, and smoking has shown a clear causal relationship to these oral diseases [61]. For example, one previous study found that current smokers were four times more likely to develop periodontitis than non-smokers [62]. Furthermore, current smokers were likely to show unhealthy behavior and negative attitudes towards general health because they were more likely to report oral health problems and less likely to use dental services, likely preventing the early diagnosis of their oral health problems [22,30].

Participation in leisure activities was associated with better self-reported oral health, a finding that was consistent with previous literature on other health outcomes [63,64,65]. Regular engagement in leisure activities could help stretch muscles and, in turn, maintain or improve different levels of physical function abilities, challenge mental abilities, enhance cognitive function, prevent social isolation, and improve self-esteem and confidence [31,66,67]. It could also help develop healthy behaviors that include more frequent oral hygiene behaviors, having dental insurance, and paying more attention to oral hygiene care [68,69]. These factors may contribute to better self-perceived oral health. Although long-lived persons are more likely to have poorer functions of ADL and physical health than their younger counterparts in general, many longevous adults could participate in daily activities independently. Thus, presumably, participation in leisure activities could improve oral health among centenarians [63].

Extensive evidence has shown that the oral hygiene behavior is positively associated with oral health outcomes [70]. Previous studies have shown that brushing teeth regularly (two times or more per day) is a good practice to maintain oral health [71]. The use of fluoride and other caries-prevention agents by toothbrushing helps prevent dental caries and periodontal disease [31]. By contrast, no or a low frequency of toothbrushing is related to a higher prevalence of diabetes mellitus and hypertension in both men and women, likely inducing more oral diseases and problems [72]. The longevity of some older adults may be related to their healthy lifestyle; thus, they may maintain more natural teeth and better self-rated oral health.

One important contribution of the present study was the finding on the moderating role of education and living arrangements in the association between oral hygiene behaviors and oral health. The findings showed that, when centenarians were formally educated or co-resided with family members, oral hygiene behaviors had a stronger positive association with better self-rated oral health and a higher likelihood of being dentate. Individuals without formal education may not be more aware of certain poor oral health symptoms and consider these symptoms a normal part of aging [73]. Evidence has shown that more educational attainment increases knowledge and information about oral health protection, improving health awareness [74]. The literature further argues that more educated older adults tend to have a larger social network, which could offer more social support and social relationships and help them develop good habits to maintain oral health [40]. Similarly, it is possible that centenarians with formal education could have a larger social network that leads to positive effects on oral health. For centenarians who co-reside with family members, we speculate that they can obtain better family support to develop and maintain good oral health behaviors [39,75]. These individuals are more likely to engage in health-promoting behaviors and obtain greater access to information and financial and emotional resources. Increased accessibility to these resources helps to reduce the consequences of stress events for these centenarians and assists them in better coping with disease and health-related risk factors. However, those who did not live with family members may not have had a supportive channel to maintain good oral health habits [40].

In addition, from the perspective of policy implications, considering the current pension model of “9073” in China, most older adults are cared for at home; institutional care is rare, less than 3% [76]. With limited social services available, especially lack of those oral health related services, centenarians’ dental care is challenging [77]. Thus, the oral health public service system could be improved from supply aspects, such as increasing resource allocation and staffing in rural, township, and community oral health service centers, and integrating dental care in the national medical insurance system.

### Limitations of the Study

We made a great effort to recruit all living centenarians in Nanjing city and to visit individually the participants to do the questionnaires. By doing so, we collected valuable, first-hand data about Chinese centenarians. However, several limitations should be noted when interpreting our findings. First, the data were of a cross-sectional design; thus, causality between oral health and health behaviors cannot be firmly established. In the future, a life-course approach that includes the time spent engaging in health behaviors and changes in sociodemographic characteristics (e.g., education, income, and co-residence) is necessary to shed further light on causality. Second, dental visits and dental insurance, two important behavior factors for older adults to maintain oral health, were not included in the current research [78,79,80,81]. However, notably, dental visits among centenarians are rare in China. Additionally, having separate dental insurance coverage is not common for older adults. In addition, given that very old persons may have few teeth, other aspects of oral health, such as cleanness of the oral cavity (particularly the soft tissue and tongue), should not be ignored because they may be particularly important to edentulous individuals. Thus, more comprehensive data in terms of clinical examinations will be required in the future to understand the relationship between oral health and longevity. Third, oral health data were self-reported possibly leading to measurement subjectivity because participants may have recall bias. Finally, the accuracy of age reporting of the participants deserves additional attention. Because the nationwide modern household registration system was implemented in the 1950s, the age of many individuals is likely inaccurate; thus, future studies should emphasize the need to certify the age of centenarians to overcome the above limitations.

## 5. Conclusions

This study is the first step in a process to provide an understanding of centenarians’ oral health status and its association with health behaviors, such as oral hygiene behaviors, leisure activities, and dietary habits. Specifically, centenarians who are non-smokers, participate in more leisure activities, and practice more oral hygiene behaviors are more likely to report better oral health. Those who eat fruits daily and practice oral hygiene behavior more frequently are more likely to be dentate. Moreover, our study illustrated the protective effect of health behaviors, particularly for centenarians with no formal education or those not living with family members. Thus, providing effective prevention and treatment throughout their life course is important in the future. Additionally, this study identified various non-disease-related factors that affect oral health, thus improving our knowledge of very old age and emphasizing the importance of lifestyle and healthy aging [82]. Given the close relationships among smoking, leisure activities, oral hygiene behaviors, eating habits, and oral health outcomes, more studies on these associations and policy implications are warranted to prevent oral diseases and promote oral health in very old persons.

## Figures and Tables

**Table 1 ijerph-18-07285-t001:** Sample by study variables.

Variables	Total Sample	Self-Reported Oral Health Status					*p*-Value	Edentulous Status		*p*-Value
		Very Poor	Poor	Fair	Good	Very Good		Lost All Natural Teeth	Having One or More Natural Teeth	
**Total N (%)**	185 (100.0)	45 (24.3)	41 (22.2)	74 (40.0)	20 (10.8)	5 (2.7)		108 (58.4)	77 (41.6)	
**Sociodemographic variables**										
Age (mean ± SD)	102.0 ± 2.0						0.293			0.412
Gender										
Women (%)	139 (75.1)	**38 (27.3)**	**26 (18.7)**	**59 (42.4)**	**14 (10.1)**	**2 (1.4)**	**0.049**	84 (60.4)	55 (39.6)	0.325
Men (%)	46 (24.9)	**7 (15.2)**	**15 (32.6)**	**15 (32.6)**	**6 (13.0)**	**3 (6.5)**		24 (52.2)	22 (47.8)	
Education										
Having no formal education (%)	128 (69.2)	35 (27.3)	28 (21.9)	52 (40.6)	12 (9.4)	1 (0.8)	0.089	**83 (64.8)**	**45 (35.2)**	**0.008**
Having formal education (%)	57 (30.8)	10 (17.5)	13 (22.8)	22 (38.6)	8 (14.0)	4 (7.0)		**25 (43.9)**	**32 (56.1)**	
Annual household income (mean ± SD)	31,459.8 ± 37,164.2						0.459			0.456
Co-residence										
Living with family members (%)	138 (74.6)	33 (23.9)	33 (23.9)	54 (39.1)	14 (10.1)	4 (2.9)	0.879	84 (60.9)	54 (39.1)	0.239
Not living with family members (%)	47 (25.4)	12 (25.5)	8 (17.0)	20 (42.6)	6 (12.8)	1 (2.1)		24 (51.1)	23 (48.9)	
**Health behavioral variables**										
Smoking										
Non-smoker (%)	149 (80.5)	34 (22.8)	30 (20.1)	64 (43.0)	16 (10.7)	5 (3.4)	0.286	86 (57.7)	63 (42.3)	0.711
Former/current smoker (%)	36 (19.5)	11 (30.6)	11 (30.6)	10 (27.8)	4 (11.1)	0 (0.0)		22 (61.1)	14 (38.9)	
Eating fruits										
Not every day (%)	132 (71.4)	35 (26.5)	31 (23.5)	52 (39.4)	10 (7.6)	4 (3.0)	0.202	82 (62.1)	50 (37.9)	0.103
Every day (%)	53 (28.6)	10 (18.9)	10 (18.9)	22 (41.5)	10 (18.9)	1 (1.9)		26 (49.1)	27 (50.9)	
Eating vegetables										
Not every day (%)	70 (37.8)	**12 (17.1)**	**26 (37.1)**	**24 (34.3)**	**5 (7.1)**	**3 (4.3)**	**0.002**	37 (52.9)	33 (47.1)	0.235
Every day (%)	115 (62.2)	**33 (28.7)**	**15 (13.0)**	**50 (43.5)**	**15 (13.0)**	**2 (1.7)**		71 (61.7)	44 (38.3)	
Leisure activities										
No participation	61 (33.0)	**18 (29.5)**	**25 (41.0)**	**12 (19.7)**	**2 (3.3)**	**4 (6.6)**	**0.000**	36 (59.0)	25 (41.0)	0.372
Participation in 1 activity	37 (20.0)	**11 (29.7)**	**7 (18.9)**	**16 (43.2)**	**3 (8.1)**	**0 (0.0)**		25 (67.6)	12 (32.4)	
Participation in 2+ activities	87 (47.0)	**16 (18.4)**	**9 (10.3)**	**46 (52.9)**	**15 (17.2)**	**1 (1.1)**		47 (54.0)	40 (46.0)	
Oral hygiene behavior										
Never (%)	75 (40.5)	22 (29.3)	21 (28.0)	26 (34.7)	6 (8.0)	0 (0.0)	0.079	**54 (72.0)**	**21 (28.0)**	**0.016**
Occasionally (%)	21 (11.4)	5 (23.8)	5 (23.8)	7 (33.3)	4 (19.0)	0 (0.0)		**9 (42.9)**	**12 (57.1)**	
Once a day (%)	55 (29.7)	8 (14.5)	11 (20.0)	29 (52.7)	5 (9.1)	2 (3.6)		**29 (52.7)**	**26 (47.3)**	
Twice a day or more (%)	34 (18.4)	10 (29.4)	4 (11.8)	12 (35.3)	5 (14.7)	3 (8.8)		**16 (47.1)**	**18 (52.9)**	
**Health status**										
Activities of daily living										
ADL dependent (%)	104 (56.2)	**26 (57.8)**	**29 (70.7)**	**41 (55.4)**	**5 (25.0)**	**3 (60.0)**	**0.021**	63 (58.3)	41 (53.2)	0.492
ADL independent (%)	81 (43.8)	**19 (42.2)**	**12 (29.3)**	**33 (44.6)**	**15 (75.0)**	**2 (40.0)**		45 (41.7)	36 (46.8)	
Cognitive function	9.45 ± 9.81						0.171			0.226
Chronic disease conditions										
No chronic disease (%)	108 (58.4)	26 (24.1)	23 (21.3)	45 (41.7)	10 (9.3)	4 (3.7)	0.770	65 (60.2)	43 (42.2)	0.555
At least one chronic disease (%)	77 (41.6)	19 (24.7)	18 (23.4)	29 (37.7)	10 (13.0)	1 (1.3)		43 (55.8)	34 (44.2)	

Note: The added percentages may not be equal to 100% due to rounding. Variables in bold means significant differences between groups (Chi-square test at *p* < 0.05).

**Table 2 ijerph-18-07285-t002:** Ordinal logistic regression results for self-reported oral health status among longevous persons (*n* = 185).

Variables	Model 1	Model 2	Model 3	Model 4
	Odds Ratio (95% CI)	Odds Ratio (95% CI)	Odds Ratio (95% CI)	Odds Ratio (95% CI)
**Sociodemographic characteristics**				
Age	1.10 (0.96, 1.26)	1.10 (0.96, 1.27)	1.10 (0.95, 1.26)	1.10 (0.95, 1.26)
Men	1.30 (0.68, 2.50)	2.43 (1.11, 5.33) *	2.41 (1.09, 5.32) *	2.66 (1.20, 5.90) *
Having formal education	1.92 (1.02, 3.62) *	1.42 (0.73, 2.74)	1.48 (0.76, 2.90)	0.68 (0.27, 1.72)
Annual household income	1.00 (0.99, 1.00)	1.00 (0.99, 1.00)	1.00 (0.99, 1.00)	1.00 (0.99, 1.00)
Living with family members	0.99 (0.54, 1.81)	0.93 (0.49, 1.76)	0.90 (0.47, 1.72)	1.38 (0.54, 3.57)
**Health behaviors**				
Former/current smoker		0.32 (0.15, 0.73) **	0.31 (0.14, 0.71) **	0.27 (0.12, 0.62) **
Eating fruits every day		1.59 (0.84, 3.02)	1.65 (0.86, 3.17)	1.41 (0.72, 2.74)
Eating vegetables every day		0.75 (0.41, 1.38)	0.71 (0.38, 1.31)	0.60 (0.32, 1.13)
Leisure activities				
Participation in 1 activity (Ref. no participation)		1.30 (0.58, 2.94)	1.29 (0.56, 2.94)	1.50 (0.65, 3.46)
Participation in 2+ activities (Ref. no participation)		2.93 (1.52, 5.65) **	2.50 (1.20, 5.21) *	2.61 (1.25, 5.45) **
Oral hygiene behavior		1.29 (1.01, 1.64) *	1.29 (1.01, 1.64) *	1.34 (0.78, 2.30)
**Health status**				
ADL dependent			0.73 (0.39, 1.37)	0.63 (0.33, 1.20)
Cognitive function			1.00 (0.97, 1.04)	1.01 (0.97, 1.04)
At least one chronic disease			0.79 (0.45, 1.38)	0.73 (0.41, 1.31)
Oral hygiene behavior × having formal education				1.90 (1.11, 3.27) *
Oral hygiene behavior × living with family members				0.73 (0.41, 1.29)

**p* < 0.05; ** *p* < 0.01.

**Table 3 ijerph-18-07285-t003:** Logistic regression analysis results for edentulous status among longevous persons (*n* = 185).

Variables	Model 1	Model 2	Model 3	Model 4
	Odds Ratio (95% CI)	Odds Ratio (95% CI)	Odds Ratio (95% CI)	Odds Ratio (95% CI)
**Sociodemographic characteristics**				
Age	1.00 (0.86, 1.16)	1.01 (0.86, 1.18)	1.01 (0.86, 1.18)	0.98 (0.83, 1.15)
Men	1.04 (0.50, 2.18)	1.35 (0.56, 3.24)	1.28 (0.52, 3.12)	1.22 (0.49, 3.06)
Having formal education	2.33 (1.15, 4.71) *	1.86 (0.87, 3.95)	1.89 (0.87, 4.08)	1.33 (0.44, 3.99)
Annual household income	1.00 (0.99, 1.00)	1.00 (0.99, 1.00)	0.99 (0.99, 1.00)	0.99 (0.99, 1.00)
Living with family members	0.74 (0.37, 1.47)	0.58 (0.28, 1.22)	0.59 (0.27, 1.26)	0.23 (0.07, 0.72) *
**Health behaviors**				
Former/current smoker		0.70 (0.29, 1.73)	0.69 (0.27, 1.72)	0.84 (0.32, 2.17)
Eating fruits every day		2.18 (1.03, 4.60) *	2.10 (0.98, 4.50)	2.21 (0.99, 4.91)
Eating vegetables every day		0.49 (0.24, 0.99) *	0.48 (0.23, 1.00) *	0.44 (0.21, 0.95) *
Leisure activities				
Participation in 1 activity (Ref. no participation)		0.70 (0.27, 1.82)	0.69 (0.26, 1.84)	0.67 (0.25, 1.83)
Participation in 2+ activities (Ref. no participation)		1.01 (0.48, 2.12)	0.75 (0.33, 1.73)	0.76 (0.32, 1.78)
Oral hygiene behavior		1.51 (1.14, 2.01) **	1.53 (1.15, 2.04) **	0.78 (0.42, 1.45)
**Health status**				
ADL dependent			0.84 (0.41, 1.73)	0.80 (0.38, 1.68)
Cognitive function			1.03 (0.99, 1.07)	1.03 (0.99, 1.07)
At least one chronic disease			0.89 (0.46, 1.73)	1.03 (0.52, 2.04)
Oral hygiene behavior × having formal education				1.38 (0.73, 2.61)
Oral hygiene behavior × living with family members				2.17 (1.10, 4.26) *

* *p* < 0.05; ** *p* < 0.01.

## Data Availability

The data that support the findings of this study are available from the corresponding author, upon reasonable request.

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
