# Peer review of "Health Behaviors and Self-Reported Oral Health among Centenarians in Nanjing, China: A Cross-Sectional Study"

_ijerph, 2021, doi:10.3390/ijerph18147285_

Round 1

Reviewer 1 Report

I have read with great interest the manuscript "Health Behaviors and Self-reported Oral Health among Centenarians in Nanjing, China: A Cross-sectional Study". The topic of the study has a quite relatively novelty in the field, even more in China, with the limitations related to access to this population. The introduction is well-written, with appropriate references and directing the text to the objectives which are clearly stated. The methods are clearly described and are adequate for this type of study, based on epidemiological data and oral health status. The results are clearly expressed and the tables are easy to read, resuming all the variables and covariables included in the M&M. I could not find any mistake or identify any better way to express the results (see minor comments). The discussion has a good flow with correct associated references comparing previous results in aging patients. The limitations section are correct but I suggest to the authors express also the efforts to recruit this sample size and to visit individually the participants to do the questionaries.

Comments

  • Exclusion criteria must be specified
  • Sample size calculation and/or potency must be included based on the primary outcome and previous similar results
  • Specify the % of participation
  • Explain the meaning of ADL
  • Please indicate the Ethical committee reference number for the matrix study
  • Inform and justify the reason of the development of ordinal and logistic regression for the same variables and objectives. 

Author Response

Reviewer 1:

I have read with great interest the manuscript "Health Behaviors and Self-reported Oral Health among Centenarians in Nanjing, China: A Cross-sectional Study". The topic of the study has a quite relatively novelty in the field, even more in China, with the limitations related to access to this population. The introduction is well-written, with appropriate references and directing the text to the objectives which are clearly stated. The methods are clearly described and are adequate for this type of study, based on epidemiological data and oral health status. The results are clearly expressed and the tables are easy to read, resuming all the variables and covariables included in the M&M. I could not find any mistake or identify any better way to express the results (see minor comments). The discussion has a good flow with correct associated references comparing previous results in aging patients. The limitations section are correct but I suggest to the authors express also the efforts to recruit this sample size and to visit individually the participants to do the questionaries.

Response: Thanks for reviewer’s suggestion. We added the following information in the revision:

“We made much effort to recruit all living centenarians in Nanjing city and to visit individually the participants to do the questionnaires. By doing so, we collected valuable first-hand data about Chinese centenarians.” (4.1 Limitations of the study, Line 86-88, Page 10)

Comments:

  1. Exclusion criteria must be specified

Response:We thank the reviewer’s helpful comments. We added the exclusion criteria in the 2.1. sampling and recruitment section in revision:

“During the data collection period, 42 individuals lost contact because of their death or migration and 48 individuals refused to participate due to poor health and/or severe cognitive impairment.” (2.1 Sampling and Recruitment, Line 108-111, page 3)

  1. Sample size calculation and/or potency must be included based on the primary outcome and previous similar results

Response:We appreciate the important point that the reviewer raised regarding the calculation of sample size. We did not conduct a sample size calculation for the study as this is a census of the centenarians who lived in the city of Nanjing at the time of survey. The list of centenarians was obtained from the Commission on Aging of Nanjing (CAN), which is based on the household registration system. CAN releases the list of centenarians very year as a mandate for centenarians’ welfare program. Among 275 centenarians on the list, 42 individuals lost contact because of their death or migration and 48 individuals refused to participate. NCS thus eventually interviewed 185 centenarians whose ages on the household booklet were 100 or older at the time of survey.

The sample size of NCS exceeded the minimum requirement (n) for the following four statistical cases that are commonly used as criteria for comparisons between groups:

(1) n = 128, with a power of 80% and alpha = 0.5 (95% of confidence interval) and a medium effective size of 0.5

(2) n = 172, with a power of 80% and alpha = 0.5 (95% of confidence interval) and a medium effective size of 0.2

(3) n = 151, with a prevalence rate of 50%, margin of error of 8% under the 95% confidence interval.

(4) n = 184, with 95% confidence interval and 5% of margin of error.

In the literature, many studies had smaller sample sizes. For example, Jopp and Rott (2006) used 56 centenarians, Jopp et al. (2017) used 151 American adults aged 18-99, and Cheung and Lau (2016) used 120 persons aged 95+ in Hong Kong. All these studies have been published in top journals in the field. In sum, we are confident about our sample size that are sufficient for this research and for robustness of the results.

We added one sentence to indicate the sufficiency of sample size for robust analysis. (Lines 112-113, Page 3)

References:

Cheung KSL, Lau, BHP. (2016). Successful aging among Chinese near-centenarians and centenarians in Hong Kong: A multidimensional and interdisciplinary approach. Aging and Mental Health, 12, 1314-1326.

Jopp D, Rott C. (2006). Adaptation in very old age: Exploring the role of resources, beliefs, and attitudes for centenarians’ happiness. Psychology and Aging, 21(2), 266-280.

Jopp DS, Jung S, Dmarin AK et al. (2017). Who is your successful aging role model? Journal of Gerontology: Social Sciences, 72(2), 237-247.

  1. Specify the % of participation

Response:Per the reviewer’s request, we specified the % of participation in the sampling and recruitment section as follows:

“The final sample included 185 participants (67.3% of the total identified) in this study.” (2.1 Sampling and Recruitment, line 111-112, page 3)

  1. Explain the meaning of ADL

Response:Per the reviewer’s request, we explained the meaning of ADL at the place where it first appeared, which followed by its abbreviation: activities of daily living (ADL). (2.2.2 Health behaviors and confounders, Line 144-145, Page 3)

  1. Please indicate the Ethical committee reference number for the matrix study

Response: Per the reviewer’s request, we added the Ethical committee reference number of this study in the revision:

“In this study, all procedures involving human participants followed the ethical standards of the Institutional Review Board (or Ethics Committee) of Ginling College, Nanjing Normal University (IRB-2017KYNCS001) and/or national research committee and the 1964 Helsinki Declaration and its later amendments or comparable ethical standards.” (2.1 Sampling and Recruitment, Line 114-117, Page 3)

  1. Inform and justify the reason of the development of ordinal and logistic regression for the same variables and objectives. 

Response: We thank the reviewer’s helpful comments. In our study, we have two outcome variables: self-reported oral health status and edentulous status, we used the ordinal regression for the self-reported oral health status and logistic regression for the edentulous status, respectively. These two outcome variables are two dimensions to reflect the oral health of individual, self-reported oral health implies the subjective perception of centenarians’ oral health, and edentulous status reflects the objective condition of oral health status, which could better reveal the oral health status of centenarians and enriching the research conclusions. We also justified the reason in the revision and modified a sentence in the Statistical Analysis section:

“The outcome variables included the self-reported oral health status and edentulous status, these two measures, captures a subjective perception and objective condition of oral health for centenarians.” (2.2.1 Oral Health Outcomes, Line 123-125, Page 3)

“We applied ordinal regression models to analyze the associations between health behavior factors and the self-reported oral health status and logistic regression models to analyze the associations between health behavior factors and edentulous status” (2.3 Statistical Analysis, Lines 161-164, Page 4)

Reviewer 2 Report

The manuscript entitled „ Health Behaviors and Self-reported Oral Health among Centenarians in Nanjing, China: A Cross-sectional Study” is very interesting and original research, considering the fact that the participants were people over 100 years of age. 

Introduction points to the problem of dental care in China, which is particularly inappropriate for the elderly.  A few questions that, if there are answers, it would be good to include in the introduction.

Is there social or private nursing home care for the elderly in China? If so, are there percentages for the share of these two forms of care? Are the elderly mostly cared for by the family?

Material and methods:   there was a cross-sectional study of 185 persons at the age of 100 and more.  A survey was conducted among these people. Do the authors know what percentage of the respondents filled the questionnaire on their own (they answered the questionnaire questions on their own) and what percentage was filled in (answers provided by the guardians) by the guardians?

A study of the cognitive function of respondents was also conducted using the short validated  Chinese version of the Mini-Mental State Examination. Would it be possible to include this abbreviated version along with a description of how the study was conducted? As an additional attachment at the end of the work or if it is problematic for the Editorial Board, simply describe the study.

Results:  the test results are presented in 3 clear tables.

Among others, the patients were divided into those with edentulousness and those who reported the presence of at least one tooth in the mouth. Were the examined patients informed about wearing prosthetic restorations? Even if there was no such information, it would be good to develop this problem of dental care in China. The work deals with very important aspects, in a sense, of the lack of social dental care in elderly patients (in this case seniors over 100 years old) with a minimum income.

Just a small note about the levels of significance: there are three listed below the tables and significance was shown only for two:  p <0,05 and p <0,01 that is why there is no need to write the third P< 0,001.

Discussion: interesting, you can additionally touch on the problems that I drew attention to in the introduction

Author Response

Reviewer 2:

The manuscript entitled“Health Behaviors and Self-reported Oral Health among Centenarians in Nanjing, China: A Cross-sectional Study” is very interesting and original research, considering the fact that the participants were people over 100 years of age. 

  1. Introductionpoints to the problem of dental care in China, which is particularly inappropriate for the elderly. A few questions that, if there are answers, it would be good to include in the introduction.

Response:We appreciate the important point that the reviewer raised regarding the problem of dental care for older adults in China. We have provided some reasons for the dental care problem of older adults in terms of socioeconomic status, physical health and cultural:

“For older adults, dental care is challenging due to their lower socioeconomic status, disadvantageous physical health and socio-cultural reasons. On the one hand, limited mobility and financial ability of older adults reduce their access to dental care, and the current dental care system is undeveloped, resulting in the lack of dental care services and nursing staff in rural, township and community [44, 45]. On the other hand, Chinese older adults are less likely to use preventative dental utilization owing to traditional beliefs and a lack of oral health knowledge [46]. For example, most Chinese older adults believe that internal fire (which signals an important body energy imbalance in traditional Chinese medicine) is the cause of gum swelling and bleeding [47]. They believe gum bleeding and tooth loss are normal, and they tend to use traditional treatments, such as resting, avoiding stress, limiting hot food intake, and taking traditional herbal remedies [47]. These behaviors reduce individuals' utilization of preventative dental services [48].” (1 Introduction section, Lines 65-77, Page 2)

  1. Is there social or private nursing home care for the elderly in China? If so, are there percentages for the share of these two forms of care? Are the elderly mostly cared for by the family?

Response: Thanks for reviewer mention this issue. In China, there are indeed social and private nursing home care for the elderly. And nursing home care is largely sponsored by the government of China with contributions from some nongovernment organizations and private investors. However, the reliable data of percentages for these two forms of care are not available.

As China’s centuries-old tradition of the extended family living arrangements is in decline, Chinese older adults are increasingly reliant on a smaller sized family and social support as means of care. However, family support is still the main care arrangement for Chinese older adults (Wu et al., 2021). Now, the “9073” pension model has become the main care model in China, that is, 90 percent of the older adults are taken care of by their family members, 7 percent are provided with community home care services, and 3 percent are provided with institutional care services (Lu, 2014).

References:

Wu., B., Cohen, M., Cong, Z., Kim, K. M., Peng, C. M. (2021). Improving care for older adults in China: Development of long-term care policy and system. Research on Aging. DOI: 10.1177/0164027521990829

Lu D P. Reflection on the patterns of old-aged care in China. China Agricultural University Journal of Social Sciences Edition, 2014,31(04):56-63.

  1. Material and methods:  there was a cross-sectional study of 185 persons at the age of 100 and more. A survey was conducted among these people. Do the authors know what percentage of the respondents filled the questionnaire on their own (they answered the questionnaire questions on their own) and what percentage was filled in (answers provided by the guardians) by the guardians?

Response: We thank the reviewer’s comments. The questionnaire we used in NCS contains eight sections: cognition function, health status, demographic characteristics, personality and psychological characteristics, socioeconomic status, lifestyle, family and social support, reproductive and family history. Since the enrollment of individuals in NCS was based on the centenarians’ family members and centenarians’ response. If older adults were too frail or severely cognitively impaired, we did not enroll them. Then, during the interviews, some questions have to be answered by centenarians, whereas others can be answered by proxy differs greatly by question.

For example, centenarians responded factual/objective questions such as the health status, demographic characteristics, socioeconomic status, lifestyle, family and social support, reproductive and family history by their own or their guardians (particularly for those questions that require recall of the answers). For subjective questions such as cognitive function, personality and psychological characteristics, most were answered by the centenarians themselves (few may be answered by proxy), and the rest were left as missing. Compared to many studies focusing on very old persons, the proxy responses are relatively low (e.g., see Gu 2008).

Gu D. (2008). General Data Assessment of the Chinese Longitudinal Healthy Longevity Survey in 2002. In Zeng Y., Poston, D., Vlosky, D.A, and Gu, D. (eds.). Healthy Longevity in China: Demographic, Socioeconomic, and Psychological Dimensions (pp39-59). Dordrecht, The Netherlands: Springer Publisher.

  1. A study of the cognitive function of respondents was also conducted using the short validated Chinese version of the Mini-Mental State Examination. Would it be possible to include this abbreviated version along with a description of how the study was conducted? As an additional attachment at the end of the work or if it is problematic for the Editorial Board, simply describe the study.

Response: We thank the reviewer’s helpful comments. We used the standardized and validated MMSE version (contains 30 questions) to measure the cognitive function of centenarians. We did not use any short version for screening centenarians’ cognitive function. The word “short” we forgot to delete it when we were revising other texts. We are sorry for this confusion. And we have deleted the “Short” in our revision. (2.2.2 Health Behaviors and Confounders, Line 150, Page 4)

  1. Results: the test results are presented in 3 clear tables.

Among others, the patients were divided into those with edentulousness and those who reported the presence of at least one tooth in the mouth. Were the examined patients informed about wearing prosthetic restorations? Even if there was no such information, it would be good to develop this problem of dental care in China. The work deals with very important aspects, in a sense, of the lack of social dental care in elderly patients (in this case seniors over 100 years old) with a minimum income.

Response: We thank the reviewer’s helpful comments. Our NCS collected the information about whether centenarians wear dentures. And in our sample, there are 71 (38.4%) centenarians wear dentures. Given that 58.4% of the participants had no teeth, our study found that almost 20% of the participants didn’t wear dentures. As the reviewer noted, such information is significant to develop the oral health problems in China, we will explore this issue in future studies.

  1. Just a small note about the levels of significance: there are three listed below the tables and significance was shown only for two: p <0.05 and p <0.01 that is why there is no need to write the third P< 0.001.

Response:Per reviewer’s comment, we delete the third significance P<0.001 below the tables.

  1. Discussion: interesting, you can additionally touch on the problems that I drew attention to in the introduction

Response: We thank the reviewer’s helpful comments. We added some policy implications from the dental care problems in China to improve the oral health of older adults.

“In addition, from the perspective of policy implications, considering the current pension model of “9073” in China, most older adults are cared at home, institutional care is rare, less than 3% [73]. With limited social services available, especially lack of those oral health related services, centenarians’ dental care is challenging [74]. Thus, the oral health public service system could be improved from supply aspects, such as increasing resource allocation and staffing in rural, township, and community oral health service centers, and integrating dental care in the national medical insurance system.” (4 Discussion, Line 78-84, Page 9-10)

Round 2

Reviewer 1 Report

The authors have addressed correctly all the queries. I recommend to publish the manuscript in the present form.

Author Response

Academic Editor comments:

This study includes valuable information regarding the oral health of Chinese centenarians. There are two minor comments: 

1. Page 1. line 44-46. As for dental caries prevention, please indicate that toothbrushing with fluoride toothpaste is effective (with some citations). Mechanical brushing only is less likely to eliminate pathogenic factors and is less effective for caries prevention.

Response:We thank the editor’s helpful comment. We revised the following sentence in the revision:

“Such as toothbrushing with fluoride toothpaste is considered one of the most effective measures to prevent dental caries because it could eliminate the pathogenic factors that are associated with dental caries, gingivitis, and periodontal disease [30-34].” (1. Introduction, Line 45-48, Page 1)

New references:

  1. World Health Organization. Fluorides and oral health. Report of a WHO expert committee on oral health status and fluoride use. WHO Technical Series 846. Geneva: WHO, 1994.
  2. Petersen P E , Lennon M A . Effective use of fluorides for the prevention of dental caries in the 21st century: The WHO approach [J]. Community Dentistry & Oral Epidemiology 2010: 32(5): 319-321.

  1. I am wondering how edentate people brush their teeth. It may depend on how toothbrushing was assessed in this study, and I would suggest adding an explanation/interpretation about it.

Response:We appreciate the important point that the editor raised regarding the toothbrushing issue. In our sample, there are 108 edentate people, and 54 centenarians had reported toothbrushing behavior, among them, 28 individuals wore dentures. Thus, we have replaced the term “toothbrushing” to “oral hygiene behaviors”. We added an explanation in the revision:

“In our sample, we measured oral hygiene behaviors including toothbrushing, brushing and cleaning dentures, and other oral hygiene care, such as wiping gum surfaces with a cloth, or brushing gums. The oral hygiene behavior was coded as ordinal variable vary from 1 (never) to 4 (twice a day or more), if participants maintained any of the above behaviors, we considered them as having oral hygiene behavior.” (2.2.2 Health Behaviors and Confounders, Line 137-142, Page 3)
